# Importance of Sanitation for Stored-Product Pest Management

**DOI:** 10.3390/insects15010003

**Published:** 2023-12-21

**Authors:** Georgina V. Bingham, David W. Hagstrum

**Affiliations:** 1Department of Entomology, University of Nebraska–Lincoln, 103 Entomology Hall, Lincoln, NE 68583, USA; 2Department of Entomology, Kansas State University, 123 Waters Hall, 1603 Old Claflin Place, Manhattan, KS 66506, USA; hgstr@ksu.edu

**Keywords:** efficacy sanitation, regulatory standards, integration, insect surveys, conferences, short courses, cost–benefit analysis

## Abstract

**Simple Summary:**

The importance of sanitation in food storage areas is sometimes overlooked. However, sanitation is vitally important to ensure safe storage of food products. Without good sanitation, the stored commodities are more likely to become infested by mold, rodents, birds, and insect pests. This paper reviews the history of sanitation legislation, courses and conferences, and research on the costs, benefits and efficacy of sanitation that have led to current recommendations and regulations.

**Abstract:**

Sanitation is essential for the cost-effective pest management of stored-product insects. The Food, Drug and Cosmetic Act of 1938 led to the Food and Drug Administration (FDA) tightening regulatory standards, and many local surveys, courses and conferences were organized to prepare the industry for these new regulations. Sanitation removes insects and residual food, which may also provide shelter for insects, with heat treatments and insecticide applications. The number of insects removed by cleaning may be reduced as the number of available hiding places increases. Decreased sanitation negatively affects the efficacy of most other pest management practices, with means of 1.3- to 17-fold decreases in efficacy. The majority of studies quantifying the efficacy of sanitation have been performed on the farm storage of grain, but some studies have been conducted for grain elevators, food processing, and the marketing system. Results ranged from no effect of sanitation alone to very effective alone or with other methods. Sanitation can also reduce insect infestation prior to harvest. Some cost–benefit analyses have been conducted for sanitation.

## 1. Introduction

Cotton [1] said “The old custom of waiting for large mill insect populations to develop and then knocking them out with one massive assault was abandoned in favor of a program which would hold insect population to a low level at all times.” Early insect detection and the removal of food residues that provide food and refuges for insect populations are important for cost-effective pest management. The availability of pheromone traps has made early detection easier. Mills and Pedersen [2] characterized the importance of sanitation programs by saying, in the first sentence of the first chapter, that “No flour mill can stay in business without an effective, practical sanitation program”. Hagstrum and Subramanyam [3] cite 41 studies showing insect species and their numbers in residual insect infestations in many different types of facilities, thus establishing the potential benefits of sanitation. Improved sanitation was an important part of the methods developed for delivering very large tonnages of containerized beetle-free semi-processed tobacco exports from India [4]. Insect-resistant packaging is widely used, but package damage can limit its effectiveness [5]. The inspection of incoming ingredients and stock rotation can minimize infested commodities as sources of insects. We look at the history of regulatory efforts to improve the cleanliness of grain and grain products in the marketing system and industry response, and then we review research to quantify the efficacy of sanitation in pest-management programs.

## 2. History of Sanitation Regulations and Clean Grain Program

The Food, Drug and Cosmetic Act of 1938 authorized the Food and Drug Administration (FDA) to prevent the interstate shipment of food containing filth or prepared, packed or held under unsanitary conditions. Hagstrum and Phillips [6] summarize the regulatory activity of the FDA, saying “Inspection programs began with foods eaten without further cooking such as those sold by bakeries” [7]. Flour mills were next. Based on studies of wheat and wheat flour [8] as well as corn and corn meal [9]), the “Clean Grain Program” was initiated. Another program to inspect local elevators began in 1952. Starting on January 1, 1955, grain with two or more rodent hairs or 2% or more by weight insect-damaged kernels was seized and released only for sale as animal feed [7]. In July 1956, the tolerances for seizures were tightened to one or more rodent hairs or 1% or more by weight insect-damaged kernels. The grain sanitation program is described by Prentice [10]. In Nebraska, a state-wide sanitation survey for stored wheat was conducted to see how prepared farmers and elevator operators were for the new regulations [11,12]. Additional information on the insect species that were found during this survey are given in Roselle [13]. *Oryzaephilus surinamensis* (L.), *Tribolium confusum* Duval, *Tenebroides mauritanicus* (L.)*, Sitophilus granarius* (L.), *Tribolium castaneum* (Herbst) and *Rhyzopertha dominica* (F.) were most prominent and were more prominent in elevators than on farms. “The elevator operator’s sanitation problems are intensified by the fact that just one load of infested grain may provide a source of infestation for all bins in the elevator.” [12]. Similar surveys were conducted for grain storages in Minnesota, North Dakota, South Dakota, and Montana [14]; sanitation on farms in Central Kansas [15]; and flour mills in Kansas, Oklahoma, and Missouri [16]. Additional papers cover fumigation [17], rodent problems [18], the clean grain program [19,20] and controlling insects infesting stored corn [21]. Nine short courses and three conferences on grain sanitation were held to help reduce insect problems (Table 1). Of the 154 presentations, 27% were on sanitation, 18% were on fumigation or residual insecticides, 14% were on insects, 6% were on microbes and 3% were on rodents or birds. Other presentations were on inspection (4%), aeration (4%), insect identification (3%), khapra beetles (3%), insects and rodents of boxcar origin (2%), extension (1%), irradiation (1%), mites (1%), sanitary equipment or facility design (1%) and other topics (12%). Two of the aeration studies were on cooling grain and four were on distribution of fumigant. Many of the speakers were well known at the time, and four of the grain sanitation courses provide an attendance list.

## 3. Efficacy of Sanitation

The efficacy of sanitation has been studied alone or in combination with other methods such as heat treatment, insecticides, fumigants, insect-resistant packaging, and semi-chemical-based methods (mass trapping, attract and kill, sex pheromone permeation) for the management of stored-product insects [22]. Decreased sanitation negatively affects the efficacy of most other pest management practices with means of 1.3- to 17-fold decreases in efficacy. Early studies were published in the 1950s, at a time when sanitation workshops were being organized to help industry meet new FDA regulations, and others were published between 1981 and 2021 at a rate of 1 to 7 per decade. The facilities studied include the bag storage of rice in a simulated warehouse; bulk grain storage on farms and in elevators; feed, flour or semolina mills; confectionary or tobacco factories; pet food and grain products areas in retail stores; and a large sea-going passenger vessel. Cleaning methods ranged from sweeping with a broom, to vacuum cleaning, to pressure cleaning with water. Results ranged from no effect of sanitation alone to very effective alone or with other methods. Sanitation can also reduce insect infestation prior to harvest.

Sanitation removes insects and residual food, which may also provide shelter for insects, with heat treatments [23] and insecticide applications [24]. For more references on the benefits of food in reducing the susceptibility of insects to pesticides, see [25]. The number of insects removed by cleaning may be reduced as the number of available hiding places increases. The proper disposal of insects and residual food removed by sanitation is necessary to prevent reinfestation. When sanitation is implemented only in areas with high trap catches to reduce cost, insects from other areas of the retail store may reinfest the area cleaned. In retail stores, the inspection of packaged commodities for insect infestation and the disposal of infested commodities may be necessary to prevent reinfestation. Structural modifications of equipment and facilities may be needed to make sanitation cost-effective. Sanitation is not a one-time cleaning process; it requires repeated cleaning.

The majority of studies on sanitation efficacy have been done with farm storage of grain. In Australia, cleaning the header of combines reduced residual insect infestations (mostly *T. castaneum*, *R. dominica* and *Sitophilus oryzae* (L.)) by 96%, from a mean of 2142 to a mean of 77, thus reducing the number of insects infesting newly harvested grain [26]. In Arkansas, the sanitation of empty bins followed by insecticide spray was effective in 18 of 21 bins of stored rice [27]. In South Dakota, stored oats bins cleaned with a vacuum cleaner before filling had significantly lower average insect-density levels than those cleaned with a scoop shovel or broom [28]. In England, pressure cleaning made a significant contribution to the effectiveness of the subsequent residual insecticide treatment against *O. surinamensis* [29]. In Australia, sanitation by itself had little effect on *O. surinamensis* infestation levels, but good hygiene improved the efficacy of protectant treatment [30]. In Kenya, nine out of ten farmers cleaned their stores before introducing the new harvest, but only half cleaned their stores during the course of storage [31]. Sweeping was the preferred method of cleaning and almost 20% of farmers, in addition to sweeping, mopped, or dusted the stores. High hygiene scores correlated significantly with lower losses. Co-storage with stover or animal feed (29%), old storage containers (41%), farm implements (30%) and other crops (65%) was common and increased losses by 2.8 percentage points. Recycling old storage bags (40%) was common, and many of these were disinfested by treating them with chemicals (53.2%), exposing them to the sun (17.7%), or dipping them in hot (19.4%) or cold (9.7%) water. 

Grain cooling early in the storage season resulted in the insect populations being controlled when adequate sanitation was practiced [32]. After this study was complete and the grain had been moved, the bins were cleaned thoroughly before cold winter weather arrived. Three weeks before the harvest, bins were recleaned and the bin bottoms were fumigated. After the bins were filled with newly harvested wheat, all grain was cooled immediately by controlled aeration. Immediately after harvest, the mean number of insects per trap was reduced to 1.3 (compared to 21 the previous year). Cleaning the empty bins at the elevator before refilling with newly-harvested wheat resulted in a significantly reduced density of pest insects in discharge spouts later, and the effect lasted at least 12 weeks after filling [33]. Pest insects observed in 41.7% of the 1575 residual grain samples from boot pits, dump pits, headhouses, rail areas, and tunnels in nine grain elevators in Kansas show the magnitude of the sanitation problem [34]. Triplehorn [35] observed a direct relationship between sanitation and insect populations. Opit et al. [36] concluded that the heat disinfestation of empty bins is more effective when used in conjunction with sanitation and cleaning procedures.

A few studies have quantified the efficacy of sanitation in food processing facilities. In USA flour mills, vacuum cleaning alone may reduce the insect population for several months when it is impractical to close down for spot fumigation [37]. In Thailand, five species of pests and four species of their natural enemies were found in bags of paddy, brown and milled rice in both the cleaned and uncleaned rooms, with more insects in the milled rice than in paddy or brown rice [38]. Cleaning the storage premises by sweeping the ceiling, walls and the surfaces of the rice bags to remove residues, dust and spider webs, just after the rice bags were brought into the room at the beginning of the experiment and once a month after sampling, decreased grain losses on all the types of rice. In Kansas flour mills, egg mortality decreased linearly with an increase in flour depth, whereas that of adults decreased exponentially during heat treatment [39]. In Indiana flour mills, the highest sanitation levels achieved the longest rebound time and thus received the maximum fumigation benefit [40]. Facilities that had poor sanitation practices rebounded very quickly, sometimes within months, to pre-fumigation levels. Sanitation, a preventative measure, may improve the effectiveness and reliability of other control methods and reduce the cost of pest management for food processors when using these methods [41]. Bioassay results suggested an equal level of effectiveness of fumigation for two facilities, but monitoring data suggested that the cleaner facility (Facility A) had slower pest population rebound rates than the less-clean facility (Facility B). A comparison of pest control costs in each facility revealed that Facility A spent less than Facility B on pest control, as predicted, but also spent less on sanitation. These findings appear to correlate with the early results of an online survey that seeks a broader perspective of industry trends. One feed mill had a more rigorous sanitation schedule than the others, and traps placed in this mill captured very few insects [42]. Residue accumulation was directly proportional to moth-pest presence in pet food mills [43]. Hand sweeping was more effective than automatic sweeping, which left residues in corners.

For a confectionery factory in Australia, the number of *Ephestia cautella* (Walker) caught in traps was inversely correlated with the level of sanitation in a room and the distance from a room with two infested chocolate refining machines [44]. Similarly, after the removal of an infested machine from an Italian confectionary factory in May, *E. cautella* trap captures decreased significantly [45]. Abnormally high pheromone trap catches in a cigarette factory ended when several hidden tobacco dust accumulations were removed [46]. When Trematerra and Gentile [47] focused sanitation primarily on the milling areas of each floor with the highest local infestation, and the level of insect pests was kept low compared to the level observed in the same months of the previous years. Inspections and the redistribution of cleaning from areas in a cigarette factory with redundant or excessive cleaning to areas missed eliminated unacceptable infestations [46]. 

Some studies on the efficacy of sanitation have been performed in retail facilities. In Canada, *Tribolium destructor* (Uyttenboogaart) was detected via visual inspection but not by trapping on three decks of a passenger vessel where food was processed and consumed [48]. Numbers of adults and larvae were reduced but *T. destructor* was not fully eradicated by cleaning and insecticide spray. In Kansas retail stores, sanitation was conducted in areas under the shelves where birdseed, dry dog food, dry cat food, and bulk-stored pet foods were displayed, and trap captures were consistently high [49]. Sanitation included sweeping and vacuuming spillage under kick plates, dust and dirt on floors, and cleaning shelves with wild birdseed and small-pet animal food products. Sanitation also included discarding the bulk-stored food products because they were infested. A total of 21 person-hours was spent performing sanitation in each store. According to store managers, the sanitation that was performed was considerably more thorough than their routine daily sanitation. The weevils, *Sitophilus* spp.; drug store beetle, *Stegobium paniceum* (L.); and red flour beetle, *Tribolium castaneum* (Herbst) in two retail pet stores in Kansas, USA, were sampled with pitfall traps on five separate occasions before and four separate occasions after thorough sanitation in areas with high trap captures. Captures of *Sitophilus* spp. in store 1 and *T. castaneum* in store 2 increased immediately after sanitation, but subsequently, they were similar to levels before sanitation, whereas captures of *S. paniceum* in store 1 and *Sitophilus* spp. in store 2 were unaffected by sanitation. Roesli [50] reported that sanitation and pesticide applications were effective in reducing beetle numbers in retail stores, but not *Plodia interpunctella* (Hübner).

Sixty species infest commodities in the field and reproduce during storage [51,52]. Another 137 insect species infest commodities in the field in sufficient numbers to be a problem requiring pest management during storage, but they do not reproduce during storage. For many of these 197 species, sanitation can reduce insect infestations in the field prior to harvest. Winter sanitation reduced the number of insecticide applications required to maintain nut damage at low levels [53]. Orchard sanitation involves the removal of nuts remaining in trees following harvest by mechanical treeshakers or hand poling and destroying the nuts on the ground. Postharvest cull tubers and volunteer plants, as food resources for overwintering *Phthorimaea operculella* (Zeller) in commercial potato fields, and sanitation methods for reducing such sources are discussed by Mbata et al. [54] and Shelton and Wyman [55]. Sanitation to reduce these food resources can ultimately reduce the infestation levels in stored potatoes. The removal of volunteer and alternate hosts may also reduce the pre-harvest infestation of commodities by other stored-product insect pests and subsequent insect problems during storage. 

## 4. Cost of Sanitation

There are few examples of the cost–benefit analysis of sanitation, but these illustrate the potential value of the calculations. Interviews of eight corn producers in Indiana and Illinois showed that their sanitation cost averaged USD 21.40 per grain bin, USD 13.38 for the bin surroundings, USD 26.95 per grain dryer, USD 17.76 per auger cleaning, USD 14.23 per combine cleaning, and USD 10.35 per truck cleaning [56]. Cleaning was performed with a broom, a brush, compressed air, or by flushing with grain, and sometimes spraying with insecticides. For 36,000 bushels, the cleaning cost is less than USD 0.001 per bushel, while benefits from avoided discounts can range from USD 0.10 to USD 0.85 per bushel. Simulations showed that boot sanitation (cleanout) about every 30 days avoided costly grain discounts from insect commingling [57]. The annual cost of cleaning and spot fumigation of two flour mills with 2500 sack capacity was USD 962.00 and USD 1011.40 [37]. For flour mills in 1986, the cost of sanitation varied from USD 0.147 to USD 0.334 per 45.36 kg (100 lb) of flour milled [58]. Some of the most extensive cost–benefit analyses have been conducted for winter sanitation to remove nuts left in trees or on the ground after harvest [59,60]. The cost of winter nut removal is roughly equivalent to that of a single insecticide application. Generally, the benefits of sanitation greatly exceed the costs. Gordon et al. [61] determined how frequently sanitation has been adopted by California nut farmers and how important cost was in this decision.

## 5. Conclusions

The movement from “the old custom of waiting for large mill insect populations to develop and then knocking them out with one massive assault was abandoned in favor of a program which would hold insect population to a low level at all times” reported by Cotton in 1964 [1] has required the effort of many people. Improvements in sanitation programs have played a large role in this effort to improve food quality. The efficacy of sanitation methods has been the subject of several studies, the efficacy of sanitation has been part of many other studies, and more studies are needed. We need to guard against complacency at a time when we are losing many laboratories whose mission is stored-product protection [62]. 

## Figures and Tables

**Table 1 insects-15-00003-t001:** Nine short courses and three conferences on grain sanitation between 1954 and 1968.

Course or Conference ^1^	Presentations: Author, Year, Title, Pages
Anonymous. 1954. Grain sanitation. A synopsis of lectures in the first. agricultural short course on grain sanitation, Pullman, Washington, 5–6 October 1954. State College of Washington.	Brannon, David H. 1954. The extension entomologist and the grain sanitation program. pp. 30–31. Cotton, R. T. 1954. Grain sanitation. pp. 13–18. Harwood, Robert F. 1954. Anatomy of insects and their conservation of water. pp. 3–4. Johnson, Carl A. 1954. Insect growth, metamorphosis, and habits. pp. 1–2. Telford, H. S. 1954. Insecticides. pp. 5–12. Walker, David. 1954. Control of grain insects in the Pacific Northwest. pp. 26–29. Winburn, T. F. 1954. Fumigants and protectants for controlling insects in stored grain. pp. 19–25. ____________________________________________________________________
Anonymous. 1955. Grain sanitation. A synopsis of lectures in the second agricultural short course on grain sanitation, Pullman, Washington, 12–14 October 1955. State College of Washington.	Bishop, Guy 1955. Field infestation by stored grain insects. pp. 12–13. Bishop, Guy 1955. Insect Identification: The “honor roll”. p. 30. Holton, C. S. 1955. The life cycle of wheat smut. pp. 14–15. Hudson, George 1955. E. Birds and mammals invading grain elevators. pp. 25–26. Padget, L. J. 1955. The Khapra beetle: A situation report. (see ARS 22-17) Saxton, Jean P. 1955. The practical problems of rodent control in elevators. pp. 27–28 Smith, Howard 1955. The Idaho stored grain insect research project. pp. 23–24. Stallcop, Pete 1955. The food and drug administration’s activities in the grain sanitation program. p. 29. Wagner, George B. 1955. How can the consumer be assured of clean food. pp. 5–10. Walker, David W. 1955. Insect control suggestions. (see Wash. Agr. Exp. Sta. Circular 275) Walker, David W. 1955. How to sample for insects. (see Wash. Agr. Exp. Sta. Circular 275) Wilbur, D. A. 1955. Effective use of grain fumigants. pp. 16–22.
Anonymous. 1956. Grain sanitation. A synopsis of lectures in the third agricultural short course on grain sanitation, Moscow, Idaho, 24–25 October 1956. University of Idaho.	____________________________________________________________________ Hefti, Roy E. 1956. Temperature-determining equipment. pp. 8–12. Krantz, G. W. 1956. Stored-grain insect research in Oregon. pp. 16–17. Romig, Glen R. 1956. Milling problems arising from grain contamination. pp. 18–20. Smith, H. W. 1956. Problems of moisture and temperature determination in grain bins. pp. 13–15. Walkden, H. H. 1956. Recent developments in stored grain insect control. pp. 1–3. Walker, David W, 1956. Stored grain insect-control research in Washington. pp. 6–7. Williams, R. T. 1956. Use of aeration systems for the distribution of fumigants. pp. 4–5. Characteristics used in laboratory identification of stored grain beetles. pp. 21–23. Grain sanitation short course attendance. pp. 24–25. _____________________________________________________________________
Anonymous. 1957. Grain sanitation. A synopsis of lectures of the fourth agricultural short course on grain sanitation, Pendleton, Oregon, 20–21 November 1957. Oregon State University Press.	Busdicker, H. B. 1957. The khapra beetle program. p. 49-. Jones, M. P. 1957. Clean grain program. pp. 4–10. Klepser, George E. 1957. Aeration fumigation systems (Panel report). pp. 31–34. O’Brien, J. F. 1957. Stored-grain insect research in Oregon. pp. 46–48. Reed, Roy B. 1957. Some observations on aeration of grain in commercial storage—facilities of the Touchet Valley Grain Growers, Inc. (Panel report). pp. 35–36. Roberts, J. 1957. Aeration of wheat by mechanical means. (Panel report). pp. 37–42. Romig, H. W. 1957. Milling company problems—Birds, rodents and insects. pp. 26–27. Smith, H. W. 1957. Research on stored-grain insects in Idaho. pp. 23–25. Walker, David W. 1957. Insect control by aeration (Panel report). pp. 28–30. Walker, David. W. 1957. Research progress report on stored grain insects. pp. 43–45. Winburn, T. F. 1957. Grain fumigants and residual spray. pp. 11–22.
Anonymous. 1958. Grain sanitation. A synopsis of lectures of the fifth agricultural short course, Pullman, Washington, 28–29 October 1958. Washington State University.	Gray, H. E. 1958. Grain fumigants, their use and misuse. pp. 33–43. Gray, H. E. 1958. Some factors in the control of grain pests. pp. 10–26. Hardwood, Robert F. 1958. Hazards of seed treatment of control of wire worms and other insects. pp. 52–53. _____________________________________________________________________ Kedzior, John A. 1958. The grain sanitation program of the Food and Drug Administration. pp. 1–4. Klepser, George. 1958. Movement of air and fumigants through bulk grains. pp. 44–47. Krantz, G. W. 1958. Research in progress in Oregon. pp. 8–9. Meek, LeRoy. 1958. Washington state food inspection as it applies to the grain sanitation program. pp. 5–7. Purdy, Henry. 1958. Seed treatments to prevent smut, why they present a hazard. p. 51. Shaw, C. Gardner. 1958. Grain molds and their control. pp. 30–32. Smith, Howard W. 1958. Stored grain insects research in Idaho. pp. 27–29. Walker, David. 1958. Specific fumigation problems in the Pacific Northwest. pp. 48–50. Grain sanitation short course attendance. pp. 54–56. _________________________________________________________________
Anonymous. 1959. Grain sanitation. A synopsis of lectures in the sixth agricultural short course on grain sanitation, Moscow, Idaho, October 27–28, 1959. University of Idaho.	Fenwick, Harry S. 1959. Cereal disease problems. Gold, Roger 1959. pp. 37–40. Grimes, Kester B. 1959. An elevator operator’s approach to grain sanitation. pp. 23–24. Gunn, J. W. and Don Whiteaker 1959. New developments in grain fumigation. Harris, Kenton L. 1959. Contamination of raw agricultural production as it affects human food. pp. 13–16. Henderson, L. S. 1959. Sanitation requirements for government stored grain. pp. 45–49. Krantz, G. W., Robert Zwick and H. W, Smith 1959. Research on stored grain insects. pp. 17–18. Smith, H. W. 1959. Idaho Experiment Station. p. 21. Udy, Doyle 1959. Rapid test for protein. pp. 25–27. Webster, Milton H. 1959. pp. 41–43. Zwick, Robert 1959. Washington Experiment Station. p. 19. Panel discussion: Grain sanitation laws of Washington, Oregon and Idaho. Leroy Meek, Leland Fife, Kenneth L. Pool and Kenton L. Harris. pp. 1–11. Panel discussion: Rodents and their control. Earl J. Larrison, Robert Gold and Milton H. Webster. pp. 31–36. Grain grading and identification of stored grain insects. Grain sanitation short course attendance. pp. 51–55. _________________________________________________________________
Anonymous. 1960. Grain sanitation. A synopsis of lectures in the seventh annual short course on grain sanitation, Pendleton, Oregon, 9–10 September 1960. Oregon State University Press.	Capizzi, J. 1960. Granary insects and their injury. pp. 41–42. Evans, K. E. 1960. Khapra beetle survey—a federal approach. pp. 33–35. Kirk, J. L. 1960. Safety measures in fumigation. pp. 23–27. Larson, F. P. 1960. Khapra beetle survey—a state approach. pp. 37–39. Morton, A. N. 1960. The clean grain program. pp. 3–5. Powelson, R. L. 1960. Seed treatment for smut. pp. 29–31. Radinovsky, S. 1960. Some recent developments in granary mite research at Oregon State College. pp. 9–11. Schoenherr, H. W. 1960. The role of elevator construction in grain sanitation. pp. 17–22. Smith, H. W. 1960. Research on stored grain insects in Idaho. pp. 7–8. Teal, Ray. 1960. Agricultural adjustment—an open discussion. Whiteaker, Don. 1960. Application techniques in fumigation and demonstration of fumigation equipment. Zwick, R. W. 1960. Research report from Washington State University. pp. 13–15. Grain sanitation short course attendance. pp. 43–46. ____________________________________________________________________
Anonymous. 1961. Grain sanitation. A synopsis of lectures of the eighth agricultural short course, Pullman, Washington, 15–16 November 1961. State College of Washington.	Adams, Cameron S. 1961. State and federal programs as they apply to country elevators Panel Discussion. pp. 15–17. Holtman, Heinrich 1961. Research report on stored grain mites. pp. 1–3. Johnson, Allen B. 1961. Looking ahead on grain sanitation—Panel Discussion. pp. 23–28. Meek, LeRoy 1961. A model grain elevator sanitation program. pp. 19–22. Read, Dick 1961. Grain sanitation in the flour mills. pp. 11–13. Sikorowski, Peter P. 1961. Relationship of fungi to stored grain insects. pp. 5–9. Grain sanitation short course attendance. pp. 29–32. Resumes of the following contributions to the program are not included in this syllabus: Blain, Harold 1961. Checklist for elevators handling food grains. Henderson, Lyman 1961. Grain sanitation problems from a national level. Henderson, Lyman and Walter Nelson 1961. State and federal programs as they apply to country elevators—Panel Discussion. Kosin, Igor 1961. Russian agriculture. Read, Dick, Roland Portman, and Jean Saxton 1961. Looking ahead on grain sanitation—Panel Discussion. Zwick, Robert and Howard Smith 1961. Progress reports on research. ____________________________________________________________________
Anonymous. 1965. Grain sanitation. A synopsis of lectures in the eleventh agricultural short course on grain sanitation, Pullman, Washington, April 7–8, 1965.	Brockington, S. F. 1966. A quality product requires sanitary ingredients. pp. 79–83. Dodson, M. 1966. What’s new in rodent and bird control around grain elevators and cereal plants. pp. 85–90. Eighme, Lloyd 1965. Hot spots and competitive effects of grain-insects. pp. 16–21. Jackson, D. S. 1965. Khapra beetle review. pp. 22–25. Logan, Charles 1965. Do’s and don’t’s of grain sanitation. pp. 2–5. Smith, Howard W. 1965. Irradiation and other “new” means of stored grain insect control. pp. 32–34. Storey, Charles L. 1965. The nature of grain fumigants. pp. 6–11. Storey, Charles L. 1965. Forced dispersion of fumigants. pp. 12–15. Telford, H. S. 1965. Grain Protectants. p. 1. Winburn, T. P. 1965. Updating grain insect’ control methods. pp. 26–31. ___________________________________________________________________
A. O. M. Sanitation Committee (ed.). 1966. Proceedings of the grain and cereal products sanitation conference. 26–28 January 1966, Kansas State University, Manhattan, Kansas.	Foltz, V. D. 1966. Microorganisms and possible toxins on grains. pp. 61–66. Gaines, J. P. 1966. The rice millers’ viewpoint of sanitation. pp. 27–31. Goossen, H. J. 1966. Selection of grain for processing. pp. 55–60. Henderson, L. S. 1966. Selection of pesticidal chemicals and application methods for product processing and storage—Insecticides and fumigants. pp. 131–140. Hurley, J. 1966. A cereal company takes a close look at sanitation. pp. 19–25. Hurtig, H. 1966. Pesticide residues on cereal grains and their milled products. pp. 99–109. Kennedy, R. 1966. Significance of sanitation in grain handling and storage. pp. 33–38. Liscombe, E. A. 1966. Insect and rodent contamination of boxcar origin. pp. 111–114. Long, K. J. 1966. A training techniques—Tape-slide presentation. pp. 41–48. Mann, D. L. 1966. What sanitation means to the baking industry. pp. 9–13. McSpadden, P. 1966. Infestation and rodent contamination of box car origin. pp. 115–119. Moses, W. R. 1966. Food and drug laws as they apply to sanitation in the grain and cereal industries. pp. 49–53. Pillsbury, P. W. 1966. A milling company’s top management viewpoint of sanitation. pp. 15–18. Stevens, S. N. 1966. Training programs for employees. pp. 39–40. Wagner, G. B. 1966. Sanitation: past, present, and future. pp. 121–129. Wagner, G. B. 1966. Airborne contaminants as they occasionally appear in food products. pp. 67–77. Wilbur, D. A. 1966. The variable factors affecting the successful application of fumigants and protectants to stored grain. pp. 91–98.
A. O. M. Sanitation Committee (ed.). 1967. Proceedings grain and cereal products sanitation conference, 16–17 February 1967. University of Minnesota, St. Paul, Minnesota.	______________________________________________________________________ Christensen, C. M. 1967. Maintaining quality of stored grains. pp. 42–43. Durham, Joe. 1967. Federal law on grain contamination. pp. 16–22. Faulkner, C. E. 1967. New aspects of bird and rodent control. pp. 90–91. Foltz, Vernon. 1967. Microorganisms and production of toxins. pp. 28–32. Harein, Phillip K. 1967. Toxicity and hazards of pesticides. pp. 92–100. Hartig, H. 1967. Pesticide residues. pp. 62–69. Henderson, Lyman. 1967. Preventing insect infestations. pp. 35–42. Hodson, A. C. 1967. Relationship between insects and microorganisms in stored grain. pp. 33–34. Laudani, H. 1967. New nonchemical insect-control methods. pp. 70–89. Lawson, Warren. 1967. Rescue and first aid measures. p. 112-. Lofgren, John. 1967. Correct storage, handling, application and disposal of pesticides. pp. 101–106. Pratt, H. D. 1967. It’s easy to identify common beetles and weevils infesting stored foods with new CDC pictorial key. pp. 23–27. Rosenthal, Abe. 1967. Minnesota law on grain contamination. pp. 11–15. Wagner, G. B. 1967. Pest, present, and future of sanitation in the grain and cereal industry. pp. 45–54. Wilbur, D. A. 1967. Variable factors affecting the successful application of fumigants and protectants to stored grain, pp. 55–61. Williams, N. H. 1967. Importance and simplicity of a sanitation program, pp. 6–10. ___________________________________________________________________
A. O. M. Sanitation Committee (ed.). 1968. Proceedings sanitation workshop grain and cereal products, 8–11 September 1968, Manhattan, Kansas.	Boyd, H. H. 1968. Flour mill sanitation problems. pp. 25–29. Champion, E. 1968. Sanitation service. pp. 137–147. Cramer, J. L. 1968. Developing and implementing a plant sanitation program. pp. 51–60. Foltz, V. D. 1968. Microorganisms and the cereal grain industry. pp. 103–107. Harein, P. K. 1968. Pesticides for use in and around grain storage mills, warehouses and grocery stores. pp. 77–87. Hibbs, A. 1968. The operative miler’s attitude toward sanitation. pp. 3–5. Jones, D. 1968. Sanitation through engineering design. pp. 99–101. Jones, D. F. 1968. Food plant sanitation inspection procedures. pp. 37–39. Kahn, J. 1968. Guidelines for use of pesticides in food plants. p. 123. Landry, L. 1968. Pest control service Part I. pp. 125–130. McCloud, T. 1968. Pest control service Part II. pp. 131–135. McSpadden, P. 1968. Contamination in transit and consumer channels. pp. 89–97. Pedersen, J. R. 1968. Packing and storage of finished products. pp. 31–35. Pedersen, J. R. 1968. Problems created by stored-products pest other than insects. pp. 71–75. Rasco, M. R. 1968. Sanitation problems and needs of the baking industry. pp. 111–119. Rutledge, J. 1968. Goals of an adequate inspection program. pp. 41–49. Rutledge, J. 1968. Possible contamination of grain during growth, harvest and storage. pp. 19–24. Strait, M. L. 1968. Concern and challenge. pp. 9–17. Wagner, G. B. 1968. Personal hygiene and habits. p. 109. Wagner, G. B. 1968. The role of the pest control operator in the American society. pp. 173–176. Walter, V. E. 1968. Communications between pest control industry and the food industry. pp. 149–171. Wilbur, D. A. 1968. Problems created by stored-products insects. pp. 61–69. Wilson, R. 1968. Bakery equipment—selection and inspection. p. 121.

^1^ Grain sanitation short courses for 1955, 1956 and 1959 are available from Washington State University Library as 633.1 W279s. The eighth in 1961 and eleventh in 1965 are available at Washington State Library, Olympia, Washington (WA378.5 Ag8p) State Depository Copy. Additional short courses offered in 1962, 1963, 1964 and 1966 have not been found. A. O. M. Sanitation Committee 1966 is available at: https://babel.hathitrust.org/cgi/pt?id=umn.31951t00020229v&view=1up&seq=5, (accessed on 12 December 2023) A. O. M. Sanitation Committee 1968 is available at Kansas State University Library and A. O. M. Sanitation Committee 1967 is available at the University of Minnesota Library.

## Data Availability

Data sharing is not applicable to this article.

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
