# Peer review of "Importance of Sanitation for Stored-Product Pest Management"

_insects, 2023, doi:10.3390/insects15010003_

Round 1

Reviewer 1 Report

Comments and Suggestions for Authors

Comments on the review

"Importance of Sanitation for Stored Product Pest Management"

 by Georgina V. Bingham and David W Hagstrum

Sanitation is an unseparated and essential part of stored product pest management. The manuscript presents an overview of the history of sanitation regulations and the Clean Grain Program, the efficacy of sanitation, and the cost-benefit analysis of sanitation for the management of post-harvest pests. The authors reviewed 61 publications in this field. Without a doubt, the review is important to members of the scientific community who investigate stored product pest management.

Here are some comments. 

The review presents a very detailed history of sanitation regulations in stored products, courses, and conferences in this field, in the 1950s and 1960s, starting in 1954 and stopping in 1968.  Why? What happened after? What is the regulation today? What are the differences throughout the world? Without the answers, the review looks incomplete.

Table 1 looks as unnecessary. Enough to sign the number of courses and conferences on sanitation and to give the main references in the text.   

Many references are old, but only a few citations from the last decade exist.

Section 3 contains many examples of sanitation efficacy. However, these reports should be given by any system. It may be done by targeted pests or by combination with some specific treatment, or any other criteria, but not as an occasional mix.

The sentence on LL523-525 is not clear.

References number 34 (LL 536-538) and 39 (LL 550-552) do not fit the paragraph. 

Citing references in the conclusion section, especially the first and the last only, seems incorrect.

After the proper correction, the review may be published.

Author Response

Please see attachment.  Thank you for the rapid reviews. 

Reviewer 2 Report

Comments and Suggestions for Authors

I think the idea of a review paper on the importance of sanitation is very good, and will be very useful to the field of postharvest pest control. The paper is therefore needed.

But I believe it needs to be better structured. As it stands, it is mainly a long list of studies, with generally a very high-level conclusion on what the study showed (or sometimes tried to show). But this lacks added value such as clearly categorizing the types of sanitation, the types of locations where sanitation has been applied (the sequence of papers cited uses this categorization, but without signposting or comments that could make it a value added), and the types of results (positive, negative or neutral): all these appear to be listed without an underlying strong narrative that would add value to a mere list of studies.

I would start with a first section describing what is meant by sanitation (it comes very late in the paper that sanitation pre-harvest is also introduced, and that is of interest, but really different from post-harvest sanitation. Also, is sanitation only sweeping or vacuuming?) I think a couple of text boxes describing actual sanitation measures from a selection of the papers reviewed would help a lot.

I did not quite see the point of the huge table 1 (only table) and that may just be because there does not seem to be a short section describing the table and reaching any conclusion about this series of training courses and events.

There are several studies that are cited in a way that did not really tell me what was of interest there. For example lines 544 to 547: I wonder what the take home message is here? Line 550 and following (paper 39) is even more puzzling for me, as there is a long description of the paper, and in the end I wonder what I should bear in mind from it?

After the paper itself is re-structured, the summary will of course need to be adapted accordingly, and should be much easier to produce.

Reviewer 3 Report

Comments and Suggestions for Authors

The authors presented a review paper about the sanitation programs and methods that were implemented during the past period, related to providing safe storage conditions for agricultural stock. From the list of courses and conferences on grain sanitation, in Table 1., the authors gave valuable information about the topics and problems of grain sanitation which were relevant for a given period.  As the paper describes the issue of sanitation relevant during the period from 10 to 70 years, I suggest that it should be visible also in the title: Importance of Sanitation for Stored Product Pest Management - a history review. Also, I have some minor suggestions:

Line 508 - Shorten the species names: T. castaneum and R. dominica since they were already mentioned before. And put the authority for Sitophilus oryzae (L.)

Line 628 - Reference number 37, should be changed to number 57.

Section 4- describes sanitation costs relevant for the period from 1976 to 2015. It would be valuable to link the estimation of costs allocated for sanitation programs today, to give the author's opinion on whether more funds are allocated for sanitation nowadays or in the past. 

Round 2

Reviewer 1 Report

Comments and Suggestions for Authors

The authors accepted some of my comments. Some others were rejected. If the authors insist on their opinion, the paper may be published.